# Classical *Borrelia* Serology Does Not Aid in the Diagnosis of Persistent Symptoms Attributed to Lyme Borreliosis: A Retrospective Cohort Study

**DOI:** 10.3390/life13051134

**Published:** 2023-05-06

**Authors:** Foekje F. Stelma, Anneleen Berende, Hadewych Ter Hofstede, Hedwig D. Vrijmoeth, Fidel Vos, Bart-Jan Kullberg

**Affiliations:** 1Department of Medical Microbiology, Radboud University Medical Center, 6525 GA Nijmegen, The Netherlands; 2Radboud Center for Infectious Diseases, 6525 GA Nijmegen, The Netherlandshadewych.terhofstede@radboudumc.nl (H.T.H.);; 3Department of Internal Medicine, Radboud University Medical Center, 6525 GA Nijmegen, The Netherlands

**Keywords:** *Borrelia*, serology, PTLDS, Lyme borreliosis, Vlse, OspA, OspC, p41

## Abstract

Objective: The diagnosis of Lyme borreliosis is based on two-tier testing using an ELISA and Western blot. About 5–10% of patients report persistent symptoms of unknown etiology after treatment, resulting in substantial difficulties in further diagnostic workup. This paper presents a study aimed at determining whether serology can differentiate between patients with persistent symptoms attributed to Lyme and other patients with Lyme borreliosis. Methods: A retrospective cohort study included 162 samples from four subgroups: patients with persistent symptoms of Lyme (PSL), early Lyme borreliosis with erythema migrans (EM), patients tested in a general practitioner setting (GP), and healthy controls (HC). ELISA, Western blots, and multiplex assays from different manufacturers were used to determine inter-test variations in PSL and to compare reactivity against *Borrelia*-specific antigens among the groups. Results: In comparing the IgG and IgM reactivity by Western blot, IgG was more often positive in the PSL group than in the GP group. The individual antigen reactivity was similar between the PSL and EM or GP groups. Inter-test agreement among the manufacturers was variable, and agreement was higher for IgG testing compared to IgM. Conclusions: Serological testing is unable to define the subgroup of patients with persistent symptoms attributed to Lyme borreliosis. Additionally, the current two-tier testing protocol shows a large variance among different manufacturers in these patients.

## 1. Introduction

Lyme borreliosis (LB) is the most common tick-borne disease in the northern hemisphere [1]. In the Netherlands, a fourfold increase was observed in the number of consultations for Lyme borreliosis in primary care between 1994 and 2018, with an increase in the estimated incidence from 39 to 148 per 100,000 inhabitants, respectively [2,3,4]. The disease is caused by several *Borrelia* species within the *Borrelia burgdorferi* sensu lato complex, with the predominant species in Europe being *B. garinii*, *B. afzelii*, and *B. burgdorferi* sensu stricto, whereas in North America, it is mainly caused by *B. burgdoferi* ss [5,6]. In humans, LB primarily affects the skin, nervous system, heart, or joints and typically manifests as a skin lesion with an expanding bull’s-eye pattern called erythema migrans (EM), which appears several days to weeks after a tick bite. The skin lesion may be accompanied by flu-like symptoms such as fever, myalgia, and headache. Later manifestations of the disease, such as neuroborreliosis, acrodermatitis chronica atrophicans (ACA), or arthritis, can present weeks or months after infection [7,8,9,10,11,12]. Unfortunately, even after the recommended antibiotic treatment, on average, 5–10% of patients report symptoms for years after infection [13], after which neuroborreliosis can reach up to 48% [14,15]. This “chronic” form of borreliosis was recently described by Steere [16]. Considering *Borrelia* diagnostics, post-treatment Lyme disease syndrome (PTLDS) and central sensitization syndrome are the most challenging entities [12]. In this paper, we refer to this patient group as people with persistent symptoms attributed to LB (PSL).

The symptoms in PSL are often non-specific and vary from chronic fatigue and musculoskeletal pain to neurocognitive difficulties, and the criteria were previously described by Wormser [17]. Although chemokine ligand 13 (CXCL13) is a recognized biomarker for acute neuroborreliosis [18], there are no specific biological markers for PSL, making it extremely difficult to define the disease. Furthermore, the etiology is unclear, and possible explanations range from persistent infection with *Borrelia* spp. to factors not related to LB [16]. 

### 1.1. Two-Tier Approach in Serological Diagnostics

Standard serological diagnosis relies on a two-tier approach, starting with a *Borrelia*-specific IgG and IgM enzyme-linked immunosorbent assay (ELISA). If the screening yields a positive or equivocal result, confirmation follows by testing the IgG and IgM reactivity by Western blotting (WB) against several *Borrelia*-specific proteins [19]. A modified two-tier approach has recently been evaluated and proven equally sensitive and specific. This modified approach consists of two consecutive *Borrelia*-specific ELISAs from different commercial brands [20,21,22].

Serological diagnostics are known to have poor positive predictive value in populations with a low pretest probability. Stage and symptom entities influence pretest probabilities and the level of false-positive serological results [3,23,24]. IgM reactivity is especially hampered by a high proportion of false-positive reactions [25]. Therefore, *Borrelia* serology should only be performed if a high pre-test probability is present. Further, laboratory specialists should only consider the serological results in combination with clinical information. Finally, clinicians should take great care not to trust alternative testing, as this might lead to misdiagnosing LB [26,27].

### 1.2. Serology in Early Lyme Borreliosis

Early borreliosis includes stage 1 and stage 2 diseases. Acute borreliosis (stage 1 disease) is often defined as symptoms developing within 30 days after exposure; however, the exact date of exposure is often difficult to pinpoint, and some scientists also define symptoms that develop within 10 weeks after exposure as early borreliosis. A large difference in the time span will influence the sensitivity of the serological test. Mounting a *Borrelia*-specific IgM response typically takes 2–4 weeks, and developing a *Borrelia*-specific IgG response may take up to two months [28,29]. Thus, the so-called “window period” in which no antibodies are detectable varies from several days to 2 months after the initial infection. When evaluating the performance of a serological test in the case of borreliosis, it is better to relate the findings to a clinical entity. Early borreliosis is best mirrored by the presence of EM, also referred to as early localized disease. In EM, the two-tier serological approach has been shown to have a limited sensitivity of 31–50% [3,29,30]. The early response is predominantly an IgM reaction with reactivity against outer surface protein C (OspC; 25 kDa), basic membrane protein A (BmpA; 39 kDa), and flagellin protein (41 kDa). Only 22% of serum samples in early localized disease have been shown to be reactive in a combination of IgG ELISA and WB, with a predominant protein reactivity against surface lipoprotein E (VlsE; 35 kDa) [31]. The intensity of the antibody response is related to the duration of the EM before antibiotic treatment and the extent of dissemination [31,32]. Because of the limited sensitivity of serology in early localized disease, it is advised to treat patients directly when EM is diagnosed and not rely on serological diagnostics [33].

In early disseminated disease (stage 2, acute neuroborreliosis, lymphocytoma, and carditis), the two-tier serology approach has a sensitivity of 63–77% [3,28,30]. In comparison to patients with EM, patients with neuroborreliosis have lower IgM optical density (OD) values and higher IgG OD values against *B. burgdorferi* antigens [34]. Cerebrospinal pleiocytosis, intrathecal antibody production of IgM and IgG isotypes, and an increased level of CXCL-13 in the cerebrospinal fluid are typically present in patients with neuroborreliosis [35].

Although IgM testing has proven to be sensitive in early *Borrelia* diagnostics, a high proportion of false-positive results has also been seen, especially if the IgM reactivity exceeded 1 month after exposure, with no evidence of IgG seroconversion [25,36]. Branda and colleagues [30] suggested an alternative two-tier approach using the IgG ELISA reactivity confirmed by one WB VlsE band. This approach reached 96% sensitivity in early disseminated disease and is, therefore, an adequate alternative to IgM testing at this disease stage and onward.

### 1.3. Serology in Late Clinical Lyme Borreliosis (ACA, Arthritis, Myocarditis, and Neuroborreliosis)

Late disseminated borreliosis (stage 3) presents as ACA, Lyme arthritis, and/or continuing neurological disease activity lasting more than 6 months [12]. The two-tier approach, including 5 of 11 IgG-positive WB bands, showed 95–100% sensitivity in this patient group [30]. To enhance specificity, a positive IgM immunoblot alone, in the absence of a positive IgG immunoblot, should not be interpreted as active LB in patients with an illness duration of more than 30 days [28].

### 1.4. Serology in Patients with Persistent Symptoms Attributed to Lyme Borreliosis (PSL)

The clinical manifestations, diagnostics, and antibiotic treatment of stages 1, 2, and 3 borreliosis are well-established. Most patients will recover, and the *Borrelia*-related inflammation will normalize. However, a small proportion of patients continue to suffer from non-specific symptoms consisting of fatigue, arthralgia, myalgia, and perceived cognitive impairment. These symptoms will typically persist in spite of antibiotic treatment. Case definitions show large variations, and diagnostic workup is difficult and often accompanied by confusion and controversy. 

The goal of this study was to determine whether there are specific antibody patterns identifying patients with PSL compared to patients with EM and healthy controls. In addition, serum samples from patients with persistent symptoms were tested using blots from different manufacturers to determine manufacturer differences within this population. Finally, we investigated if pre-treatment serology could be used to predict which PSL patient was going to benefit from antibiotic treatment or not. 

## 2. Materials and Methods

### 2.1. Population

A retrospective cohort study included 162 subjects from four population groups: (1) Patients with persisting symptoms attributed to LB after antibiotic treatment (n = 40, PSL) were recruited from the Persistent Lyme Empiric Antibiotic Study Europe study (PLEASE study) [37,38]. (2) Each PSL patient was requested to bring an age-, gender-, and geographically-matched friend with no prior history of LB. These subjects formed the healthy control group (n = 40, HC). (3) Patient samples obtained from general practices that tested positive for LB were selected from our Radboudumc biobank and anonymized (n = 41, GP). (4) For patients from a Romanian cohort study [39], serological testing was performed before the start of antibiotic treatment, on average, 19 days after developing a skin lesion (n = 41, EM).

### 2.2. Serological Testing

Between 2014 and 2016, samples from groups 1 to 3 were tested according to our standard in-house two-tier protocol using the Serion IgG and IgM ELISA (Virion/Serion GmbH, Würzburg, Germany) and Euroimmun EUROLINE-WB (Euroimmun Medizinische Labordiagnostika AG, Lübeck, Germany), according to the manufacturers’ protocol (Table 1). In addition to the standard test protocol, the cases and healthy controls from the PLEASE population, groups 1 and 2, were tested with four additional commercial Western blots and multiplex assays (Table 1). The antigen compositions of the *Borrelia* Western blots and multiplex assays are depicted in Table 2. Two of the assays were performed by the manufacturer, blinded to the patient category. The Romanian cohort, group 4, was tested in 2015 using a two-tier protocol for IgG and IgM either using ELFA (Vidas, BioMerieux, Boston, MA, USA) (n = 7/41) or ELISA (Euroimmun Medizinische Labordiagnostika AG, Lübeck, Germany) as a screening test, which was chosen randomly based on availability. In this group, all blots were performed using Euroimmun EUROLINE-RN-AT (Euroimmun Medizinische Labordiagnostika AG, Lübeck, Germany). Serological results in all groups were defined as either being positive or negative according to the advised cut-offs by the manufacturers. Equivocal or borderline results were interpreted as negative for the analyses.

### 2.3. Antibiotic Treatment

The patients in group 1 were treated with antibiotics according to a treatment protocol described by Berende [37,38]. In short, the patients received 2000 mg of ceftriaxone daily for 14 days. Afterwards, a 12-week oral course of 100 mg of doxycycline, 500 mg/200 mg of clarithromycin–hydroxychloroquine, or a placebo. The subjects were asked to report beneficial or no beneficial effects from the antibiotic therapy on a standardized questionnaire (the short-form health survey SF-36), and serum samples were taken at 14 and 26 weeks after the start of treatment. Week 14 was at the end of the treatment period. Week 26 was 12 weeks after the end of the antibiotic treatment. The patients were unaware of being in either the placebo or antibiotic treatment arm.

### 2.4. Statistical Analysis

Statistical analysis was performed using SPSS version 25. Group differences were analyzed using the Chi-square test and Fisher’s exact test for the categorical data and the t-test and Kruskal–Wallis one-way analysis of variance for the continuous data. Differences with *p*-values below 0.05 were deemed significant. Inter-test agreement was determined using Cohen’s Kappa coefficient, with values of 0.41–0.60 as moderate agreement, 0.61–0.80 as substantial, and 0.81–1 as almost perfect [40].

## 3. Results

### 3.1. Subjects

A total of 162 patients were included in the study. The patient and subgroup characteristics are shown in Table 1. There were significant differences among the groups regarding age, with the highest age in the primary care group (GP) and the other three groups being of similar age. There was no significant difference in gender among the groups. 

### 3.2. Borrelia spp. Serological Testing—Manufacturer Performance in PSL Patient Group Compared to HC

Comparing the seropositivity rates in the PSL group to those in the HC, the rates of both IgG and IgM positivity were significantly higher in the patient group with persistent symptoms (PSL) (Table 3). This difference was similar for all manufacturers tested (*p*-value < 0.05). Background *Borrelia* seropositivity, which is approximately 9% (IgG) in the Netherlands, was comparable to the positive IgG rate in the HC group (between 5 and 13%, depending on which manufacturer’s test was used). Since it is standard practice to perform two-tier testing in *Borrelia* serology, the inter-test agreement was determined using Cohen’s kappa statistic (Table 4). The agreement among the IgG assays was, on average, higher than among the IgM assays. For IgG assays, substantial agreement was reached in 5 out of 15 assays, scoring almost perfect agreement, and in 10 out of 15 assays, showing substantial agreement. For the IgM assays, 11 out of 15 comparisons scored only moderate agreement.

When looking in detail at the reactivity to the individual *Borrelia* antigens in the immunoblot and multianalyte assays scoring positive in the PSL and HC groups, VlsE was the antigen most frequently showing reactivity to IgG in the PSL group across all commercial assays, with 60% positivity (120/200), followed by decorin-binding protein A (p18 or DbpA) with 32% positivity (64/200), p83/p100 protein with 31% positivity (50/160), p39 protein with 24% positivity (39/160), and p58 protein with 23% positivity (32/160). OspC/p25 was the antigen that most frequently showed reactivity to IgM in the PSL group across all commercial assays, with 37% positivity (73/200). The antigen p41 reactivity against IgG and IgM did not differ between the PSL and HC groups, as substantial proportions scored positively in both groups, with 37% (74/200) versus 33% (65/200) and 24% (48/200) versus 16% (32/200), respectively. VlsE-IgG and OspC-IgM can, therefore, be regarded as the most specific tests in differentiating between the PSL and HC groups, and p41 IgG and IgM as the least specific tests (Table 2).

### 3.3. Comparison of Borrelia Serology among Patients with Persistent Symptoms Attributed to Lyme Borreliosis (PSL), Healthy Controls (HC), and Patients in Primary Care (GP)

The samples from the PSL, HC, and GP groups were tested using similar manufacturers (Table 1) and, therefore, comparable. The percentage of positive serology (IgG and IgM) was significantly higher in the patients in the PSL and GP groups compared to those in the HC group, as shown in Table 5. Between the PSL and GP groups, there was no significant difference in the positive results, neither for the IgG nor the IgM serology.

When studying the different antigens in the Euroimmun EUROLINE-WB in more detail (Figure 1), no significant differences were found between the PSL and GP groups. However, clear differences were found when comparing the reactivity to various *Borrelia*-specific proteins between the PSL and GP groups and the HC group. However, although not significant, the reactivity to IgG-p31 (also called outer-surface protein A (OspA)) was seen more frequently in the HC 7/40 (18%) and GP 8/40 (18%) groups compared to the PSL 4/40 (10%) group (Chi square = 1.0; *p* = 0.309). Considering IgM, only 2 positive immunoblots were found in the HC group compared to 18 and 15 in the PSL and GP groups, respectively, with p25 (also called OspC) being the predominant antigen contributing to a positive result. 

### 3.4. Comparison of Borrelia Serology between Patients with Persistent Symptoms Attributed to Lyme Borreliosis (PSL) and Acute Borreliosis (EM)

There was a significant difference in the IgG seropositivity tested with Euroimmune EUROLINE–RN–AT when comparing patients with persistent and acute symptoms related to LB, showing that IgG seropositivity is more frequent in the PSL group compared to the EM group (61% versus 37%, respectively, *p* < 0.05). IgM seroreactivity was more frequent in the EM group compared to the PSL group, at 56% and 44%, respectively (*p* ≤ 0.05). Both the PSL and EM groups showed significantly more positive tests compared to the HC group (Figure 2).

### 3.5. Pre-Treatment Borrelia Serology in Patients with Persistent Symptoms Attributed to Lyme Borreliosis (PSL); Comparison between Patients with and without Response to Antibiotic Treatment

Those in the PSL group reporting beneficial effects from antibiotic therapy on a standardized questionnaire (SF-36) demonstrated equal pre-treatment seroreactivity to all *Borrelia*-specific proteins in both IgG and IgM assays, as tested by Euroimmune EUROLINE–RN–AT, when compared to patients reporting no benefits from antibiotic treatment (Figure 3). 

## 4. Discussion

This study compared the serology in patients with persistent symptoms attributed to LB (PSL) to the serology in other groups of patients with LB and healthy controls. First, this study confirms the high variability in the serological test results of different manufacturers, as has been seen by others [3,19,25,41]. Second, this investigation also confirms that the test sensitivity largely depended on the duration of the *Borrelia* infection and the disease stage [19,30]. Most importantly, this study shows that the current serological tests do not aid in defining the chronic nature of LB, as no differences in *Borrelia*-specific reactivity to various proteins could be identified when comparing the PSL group to patients with LB seen in general practice (GP). In addition, *Borrelia*-specific antigen profiles, as tested by Western blot prior to antibiotic treatment, did not predict treatment success in patients with persistent disease (PSL).

The current Dutch guidelines recommend treating patients with clear clinical signs of acute LB (erythema migrans) without additional serological confirmation due to the fact that serology is an insensitive diagnostic in acute borreliosis [32]. This practice may have caused a selection bias towards the inclusion of late borreliosis, especially in our GP population. A Dutch study aimed at describing GPs’ diagnostic behavior when suspecting LB showed that many patients were tested for borreliosis when presenting with non-specific general symptoms [42]. Therefore, patients tested for borreliosis in primary care (GP) in the Netherlands may resemble patients tested in tertiary care when presenting with persisting symptoms attributed to LB (PSL). Our study also compared antibody patterns between patients with persistent symptoms (PSL) and patients with acute borreliosis (EM). This revealed that IgG testing is more appropriate when suspecting persistent LB, while IgM is the better test when suspecting acute borreliosis (see Figure 2). Nevertheless, once a patient is known to be *Borrelia* seropositive, repeating serological diagnostics in tertiary care does not appear to contribute significantly to the diagnosis of persistent symptoms attributed to LB (Figure 1). 

Regarding the specific anti-*Borrelia* antibodies tested, our study shows that the p41 flagellin antigen in both IgG and IgM testing showed an equal frequency of reactivity in the PSL and HC groups and, therefore, does not aid in the diagnosis of *Borrelia* infection. The flagellin antigen is known to be an early marker for *Borrelia* infection [43]; however, cross-reactivity with antibodies from other spirochetes (causing syphilis or leptospirosis) was also described [43]. Interestingly, Ulvestad and colleagues, who studied the biological significance of anti-p41 IgM, also showed equal reactivity to p41-IgM in *Borrelia* in the case and control sera. They estimated that 1.5% of the general population is reactive against p41. Further, they showed that p41-positive sera could immobilize a *B. afzelii* reference strain in vitro, indicating that anti-p41 IgM may be a sign of resistance to infection rather than a sign of infection [44]. The p41 protein is still part of several Western blot assays (see Table 2). This has been recognized by manufacturers, as, for instance, its significance is only attributed one point in the test algorithm of Mikrogen RecomLine when scoring the total blot results (each positive band on a Western blot being attributed a certain score). However, since the positivity of p41 in the controls did not differ significantly from that in patients, scoring this antigen may contribute to false-positive results for both IgG and IgM in *Borrelia* diagnostic assays. 

In our study, the presence of Euroimmun EUROLINE–WB protein p31 (OspA), especially in the case of IgG testing, did not differentiate between patients with symptoms attributed to *Borrelia* infection (PSL) and healthy controls (HC; Figure 1). OspA is a spirochaetal protein that is required to infect the tick. Furthermore, OspA is expressed by spirochetes in the tick gut during the first 24 to 48 h after a tick bite, causing the (human) host immune system to be exposed only during the very first period of infection [45]. Chandra and colleagues demonstrated increased reactivity against this protein in patients with post-LB syndrome (PLDS) [46]. They suggested that this finding might be indicative of increased inflammation during the initial phase of infection in patients who, later on, develop PLDS. In our study, we observed that healthy controls were equally or slightly more frequently seroreactive against this protein. The study performed by Chandra and colleagues was performed in the U.S. It is known that *Borrelia* infections in the U.S. are predominantly of the *B. burgdorferi ss* genotype, and in Europe, they are predominantly of the *B. afzelii* or *B. garinii* genotype [5]. Moreover, U.S. *Borrelia* strains are believed to induce higher levels of Th1 immune responses, whereas strains from Europe appear to induce greater Th17-associated responses [5]. This may explain why the reactivity to OspA was more or less absent in our European PSL group. In Europe, it may be hypothesized that the presence of antibodies against this antigen is indicative of an effective immune response against *Borrelia* infection. 

Our study confirms previous observations that the IgG antibody response against VlsE is the most important marker for past *Borrelia* infection. This applies to those with acute borreliosis and to patients with persistent symptoms attributed to LB, and it was seen for all commercial brands tested. VlsE is a surface lipoprotein and is responsible for immune evasion due to continuous changes in the VlsE sequence during mammalian infection. This protein was shown to be present in the infected host from day 7 through day 21 [45]. Despite its high antigenic variation, VlsE generates a robust antibody response, and both full-length recombinant VlsE and the C6 peptide (corresponding to invariant region 6) are widely used in immunodiagnostic tests for LB [20,47]. We observed a decreased frequency of VlsE reactivity in our PSL population, which may reflect waning IgG immune responses over time [48].

Our study demonstrated reactivity against OspC (p25) IgG and IgM in about 40% of the PSL and GP populations. The proportions of seroreactivity did not differ significantly between these two patient groups. However, reactivity against OspC clearly distinguished patients with a previous *Borrelia* infection from healthy controls (HC) (Figure 1). OspC is a protein necessary for the spirochete to establish infection in a mammal by tick transmission. The production of this protein increases when infected ticks feed and shuts off within the first few weeks of mammalian infection [31,45,49]. The antibody response against OspC is an early response and is often described as being mainly of IgM origin [31]. In humans, humoral responses against the conserved regions of OspC have been shown to be related to the presence and persistence of *Borrelia*-specific symptoms and, therefore, are considered a reliable marker in serodiagnosis [31]. However, it was also shown that cross-reactivity against *Borrelia*-unrelated OspC-like proteins with similar immuno-reactive epitopes may cause cross-reactivity with *Borrelia*-specific OpsC-IgM [50]. This observation provides a cross-reactive explanation for the persistence over time of *Borrelia* OspC-IgM in patients without symptoms. Several studies have argued against testing for IgM in late LB because of the high frequency of false-positive results [23,25,30,51]. In general, an isotype switch toward IgG responses is expected after several weeks of infection. However, in *Borrelia* infections, it has been described that IgM can remain positive for up to 3 years after infection without IgG seroconversion [32]. Considering this and the positive association between OspC-IgM reactivity and symptoms attributed to LB, as described above [31], not testing for OspC-IgM in our study would have led to 14% of false-negative classifications of patients with persistent symptoms attributed to LB (PSL). 

The performance of serological tests very much depends on the stage of *Borrelia* infection in the population in which the test is being used. A study testing for *Borrelia* antibodies in patients with non-specific symptoms did not find an association between the presence of *Borrelia*-specific antibodies and the occurrence of non-specific symptoms [52]. The authors therefore concluded that *Borrelia* serology does not provide useful information in the diagnostic workup of patients with PTLDS. This may, however, be a biased observation, as the included patients were those with non-specific symptoms and not those with persistent symptoms after treatment attributed to Lyme (PSL). Looking at the evidence in studies with well-defined LB (see introduction), the most likely serological pattern in the PSL group would be low to moderate reactivity of the IgG ELISA combined with 5 IgG WB bands. However, it is possible that this patient group has different immunological responses, as we do not know what causes the symptoms. It is known that early antibiotic treatment influences the development of antibodies in some patients, rendering a group unable to develop IgG [32], and it has been shown that IgM reactivity may persist for years in others [53]. Sustained positive titers may indicate a long-term serologic memory resulting from antigen-independent polyclonal activation and the differentiation of memory B cells [48]. Considering the long-term *Borrelia* serologic memory after infection, it is not likely that the seroreactive pattern of either IgG or IgM immune responses will change over time. Therefore, we conclude that serology does not contribute to the diagnostic workup in PSL.

## 5. Conclusions

Serological test results in patients with persistent symptoms attributed to LB differed significantly among manufacturers, which can explain the variation of results among various laboratories. The current serological tests cannot be used to distinguish between patients with recent LB and patients with persistent symptoms attributed to LB. Furthermore, it is not possible to distinguish beforehand between patients who will or will not respond to antibiotic treatment through serological testing. Therefore, serological testing does not add to the algorithm for chronic LB. There are still unresolved questions, especially when considering the geographical differences in immune responsiveness and the long-term persistence of IgM in patients without Lyme-related symptoms. Additional research is needed, especially to compare the specific antibody responses over time in well-defined patient groups across the whole spectrum of LB, including patients from various geographical areas and with persistent symptoms attributed to LB. 

## Figures and Tables

**Figure 1 life-13-01134-f001:**
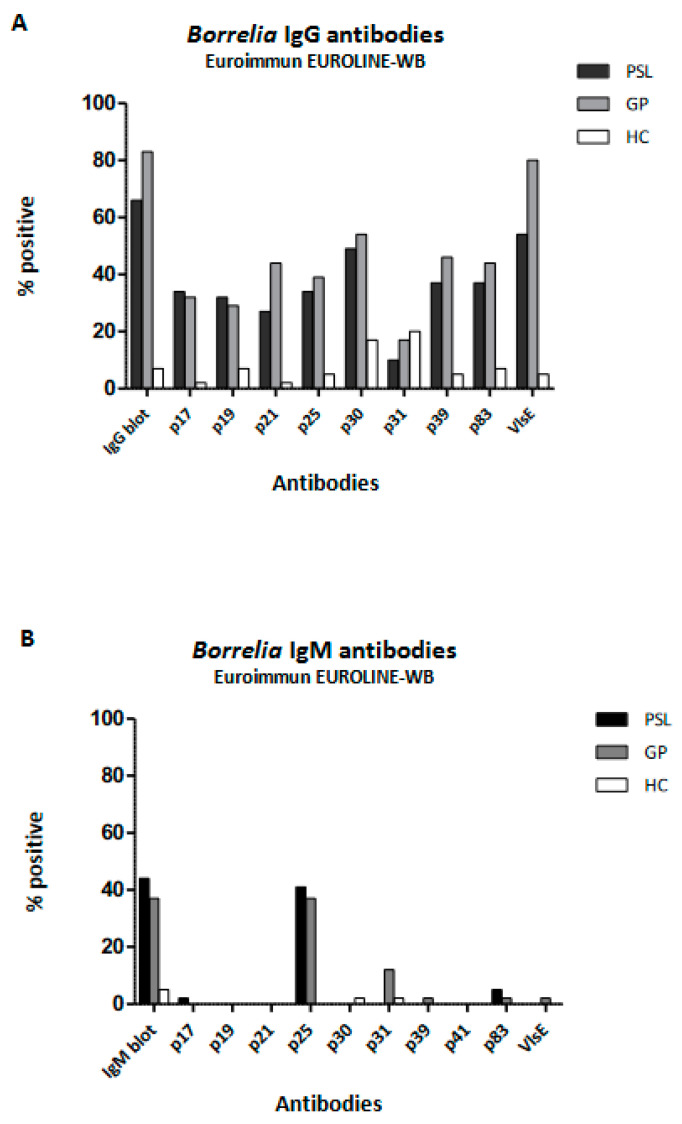
Reactivity against specific *Borrelia*-specific proteins tested with Euroimmun EUROLINE-WB. Comparison of *Borrelia* serology IgG (**A**) and IgM (**B**) among patient groups with persistent symptoms attributed to Lyme borreliosis, healthy controls, and patients in primary care. PSL—patient group with persistent symptoms attributed to Lyme borreliosis; HC—healthy controls; GP—patients in primary care.

**Figure 2 life-13-01134-f002:**
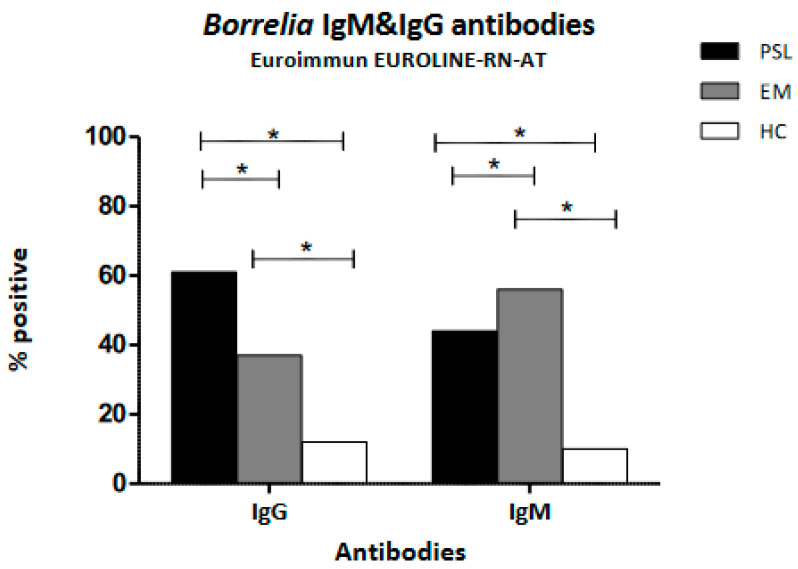
IgG and IgM *Borrelia* serology positivity as determined by Euroimmun EUROLINE–WN–AT. Comparison among patient groups with persistent symptoms, acute Lyme borreliosis as diagnosed by the presence of erythema migrans, and healthy controls. The statistical analysis used was Pearson’s Chi square test; * *p* < 0.05. PSL—patient group with persistent symptoms attributed to Lyme borreliosis; EM—patients with acute Lyme; HC—healthy controls.

**Figure 3 life-13-01134-f003:**
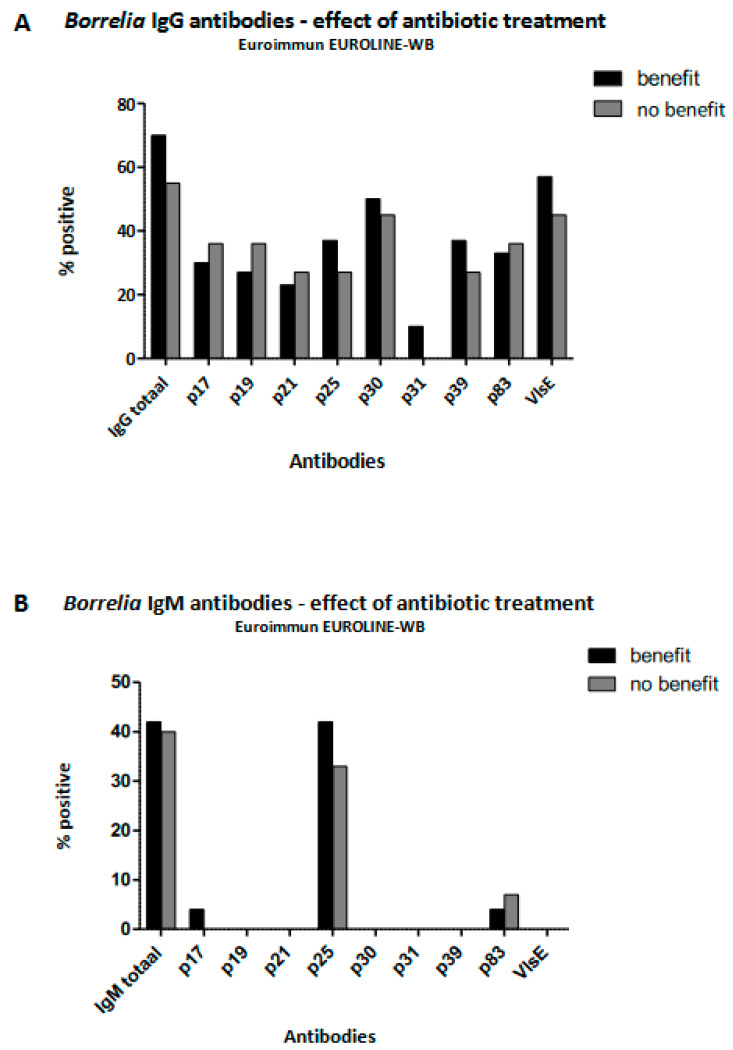
IgG and IgM *Borrelia* serologies as determined by Euroimmun EUROLINE–WN–AT in patients with persistent symptoms attributed to Lyme borreliosis. Comparison between those reporting having benefitted from antibiotic therapy and those who had not.

**Table 1 life-13-01134-t001:** Characteristics of the four different population groups investigated in this study and the commercial serological assays performed in each population group.

Group	PSL (N = 40)	HC (N = 40)	GP (N = 41)	EM (N = 41)	
Group definition	Persistent symptoms attributed to Lyme disease	Controls	Primary care	Acute Lyme (erythema migrans)	
Borrelia assays	Serion ELISA ↓ Euroimmun EUROLINE–WB Euroimmun EUROLINE–RN–AT Mikrogen RecomLine * Mikrogen RecomBead * Serion Multianalyte system ^$^	Serion ELISA ↓ Euroimmun EUROLINE–WB Euroimmun EUROLINE–RN–AT Mikrogen RecomLine Mikrogen RecomBead Serion Multianalyte system	Serion ELISA ↓ Euroimmun EUROLINE–WB	Euroimmun ELISA/BioMérieuxVidas Lyme ELFA ↓ Euroimmun EUROLINE–RN–AT	
					***p*-value**
Female gender	20 (49%)	26 (65%)	23 (56%)	25 (61%)	0.62 ^#^
Average age (years)	48	48	61	48	<0.001 **
Country	Netherlands	Netherlands	Netherlands	Romania	

Data are presented as numbers and percentages. Statistical analysis used: # = Pearson’s Chi square test, and ** = Kruskal–Wallis test. PSL—patient group with persistent symptoms attributed to LB; HC—healthy controls; GP—patients in primary care with *Borrelia* seropositivity; EM—patients with a recent clinical diagnosis of EM. * Mikrogen RecomLine and Recombead (Mikrogen GmbH, Neuried, Germany); $ Serion Multianalyte (Virion/Serion GmbH, Würzburg, Germany). Arrow ↓ indicates that an ELISA assay was performed first before the other assays depicted under.

**Table 2 life-13-01134-t002:** Comparing seroreactivity in various commercial assays to *Borrelia*-specific proteins (IgG and IgM) between patient groups with persistent symptoms attributed to Lyme borreliosis and healthy controls.

		PSL N = 40	HC N = 40	*p*-Value
VlsE				
Positive Euroimmun EUROLINE–WB	IgG	22	4	<0.001
	IgM	0	0	
Positive Euroimmun EUROLINE–RN–AT	IgG	22	3	<0.001
	IgM	0	0	
Positive Mikrogen RecomLine	IgG	26	4	<0.001
	IgM	0	2	0.358
Positive Mikrogen RecomBead	IgG	26	3	<0.001
	IgM	1	1	0.603
Positive Serion Multianalyte system	IgG	24	4	<0.001
	IgM	2	1	0.556
p17/p18/DbpA				
Positive Euroimmun EUROLINE–WB	IgG	14	1	0.002
	IgM	1	0	0.484
Positive Euroimmun EUROLINE–RN–AT	IgG	4	0	0.148
Positive Mikrogen RecomLine (*B. afzelii*)	IgG	13	2	0.007
	IgM	0	0	
Positive Mikrogen RecomBead (*B. afzelii*)	IgG	12	2	0.007
	IgM	0	0	
Positive Serion Multianalyte system	IgG	21	3	<0.001
	IgM	0	0	
p19				
Positive Euroimmun EUROLINE–WB	IgG	13	3	0.027
	IgM	0	0	
Positive Euroimmun EUROLINE–RN–AT	IgG	1	0	0.484
p20				
Positive Euroimmun EUROLINE–WB	IgG	11	1	0.01
	IgM	0	0	
Positive Euroimmun EUROLINE–RN–AT	IgG	1	0	0.390
P25/OspC				
Positive Euroimmun EUROLINE–WB	IgG	14	2	<0.001
	IgM	17	0	<0.001
	IgM	18	3	<0.001
Positive Mikrogen RecomLine (IgG positive and equivocal)	IgG	1	1	
	IgM	14	0	<0.001
Positive Mikrogen RecomBead	IgG	1	1	
	IgM	11	0	<0.001
Positive Serion Multianalyte system	IgG	5	3	0.156
	IgM	13	9	<0.001
p30				
Positive Euroimmun EUROLINE–WB	IgG	20	6	0.002
	IgM	0	1	0.360
p31/OspA				
Positive Euroimmun EUROLINE–WB	IgG	4	7	0.541
	IgM	0	1	0.06
Positive Mikrogen RecomLine	IgG	0	1	0.314
	IgM	0	0	
Positive Mikrogen RecomBead	IgG	0	0	
	IgM	0	0	
p39				
Positive Euroimmun EUROLINE–WB	IgG	15	2	<0.001
	IgM	0	0	
Positive Euroimmun EUROLINE–RN–AT	IgG	12	1	0.003
	IgM	0	0	
Positive Mikrogen RecomLine	IgG	11	2	0.006
	IgM	2	1	0.717
Positive Mikrogen RecomBead	IgG	1	1	0.603
	IgM	0	0	
p41				
Positive Euroimmun EUROLINE–WB	IgG	0	0	
	IgM	0	0	
Positive Euroimmun EUROLINE–RN–AT	IgG	39	37	0.256
	IgM	19	10	0.041
Positive Mikrogen RecomLine	IgG	33	27	0.273
	IgM	29	22	0.121
Positive Serion Multianalyte system	IgG	2	1	0.309
	IgM	0	0	
p58				
Positive Euroimmun EUROLINE–RN–AT	IgG	4	1	0.123
Positive Mikrogen RecomLine	IgG	14	2	<0.001
	IgM	0	1	0.314
Positive Mikrogen RecomBead	IgG	15	2	<0.001
	IgM	0	0	
Positive Serion Multianalyte system	IgG	2	1	0.556
	IgM	0	0	
p83/p100				
Positive Euroimmun EUROLINE–WB	IgG	15	3	0.001
	IgM	2	0	0.384
Positive Euroimmun EUROLINE–RN–AT	IgG	14	3	0.014
Positive Mikrogen RecomLine	IgG	16	2	<0.001
	IgM	2	3	0.697
Positive Mikrogen RecomBead	IgM	2	0	0.152
Positive Serion Multianalyte system	IgG	5	4	0.801
	IgM	0	0	
Others				
Positive Euroimmun EUROLINE–RN–AT Lipid	IgG	1	1	0.708
Positive Serion Multianalyte system DbpAPBr	IgG	8	2	0.103
	IgM	0	0	
Positive Serion Multianalyte system Lysate	IgG	9	2	0.074
	IgM	12	3	0.026

The data are presented as numbers. The statistical analysis used was Pearson’s Chi-square test. PSL—patient group with persistent symptoms attributed to Lyme borreliosis; HC—healthy controls.

**Table 3 life-13-01134-t003:** Comparing IgG and IgM *Borrelia* seropositivity to various commercial assays in a patient group with persistent symptoms attributed to Lyme borreliosis to seropositivity in healthy controls.

		PSL N = 40 (%)	HC N = 40 (%)	*p*-Value
Positive Serion ELISA (%)	IgG	23 (58)	2 (5)	<0.05
	IgM	16 (40)	2 (5)	<0.05
Positive Euroimmun EUROLINE–WB	IgG	27 (68)	3 (8)	<0.05
	IgM	18 (45)	2 (5)	<0.05
Positive Euroimmun EUROLINE–RN–AT	IgG	25 (63)	5 (13)	<0.05
	IgM	18 (45)	4 (10)	<0.05
Positive Mikrogen RecomLine	IgG	20 (50)	3 (8)	<0.05
	IgM	14 (35)	1 (3)	<0.05
Positive Mikrogen RecomBead	IgG	19 (48)	2 (5)	<0.05
	IgM	11 (28)	0 (0)	<0.05
Positive Serion Multianalyte system	IgG	24 (60)	4 (10)	<0.05
	IgM	14 (35)	2 (5)	<0.05

Data are presented as numbers and percentages. The statistical analysis used was Pearson’s Chi-square test. PSL—patient group with persistent symptoms attributed to Lyme borreliosis; HC—healthy controls.

**Table 4 life-13-01134-t004:** Inter-test agreement for manufacturers of IgG/IgM tests using Cohen’s Kappa coefficient.

IgG	Euroimmun EUROLINE–WB	Euroimmun EUROLINE–RN–AT	Mikrogen RecomLine	Mikrogen RecomBead	Serion MultianalyteTM System	Serion ELISA
Euroimmun, EUROLINE–WB						
Euroimmun, EUROLINE–RN–AT	0.680					
Mikrogen, *Recom*Line	0.692	0.636				
Mikrogen, *Recom*Bead	0.688	0.631	0.875			
Serion Multianalyte^TM^ System	0.676	0.622	0.685	0.679		
Serion, ELISA	0.807	0.641	0.822	0.817	0.803	
**IgM**	**Euroimmun** **EUROLINE–WB**	**Euroimmun** **EUROLINE–RN–AT**	**Mikrogen** **RecomLine**	**Mikrogen** **RecomBead**	**Serion** **MultianalyteTM System**	**Serion** **ELISA *Borrelia***
Euroimmun, EUROLINE–WB						
Euroimmun, EUROLINE–RN–AT	0.613					
Mikrogen, *Recom*Line	0.600	0.409				
Mikrogen, *Recom*Bead	0.647	0.592	0.451			
Serion, Multianalyte^TM^ System	0.643	0.589	0.400	0.690		
Serion, ELISA	0.586	0.535	0.429	0.543	0.552	

White—almost perfect agreement; light grey—substantial agreement; dark grey—moderate agreement.

**Table 5 life-13-01134-t005:** Comparing IgG and IgM *Borrelia* seropositivity among patient groups with persistent symptoms attributed to Lyme disease, healthy controls, and patients in primary care.

		PSL n = 40 (%)	HC n = 40 (%)	GP n = 41 (%)	*p*-Value
Positive Serion screening ELISA	IgG	23 (58)	2 (5)	32 (78)	<0.001 * 0.098 ^#^
	IgM	16 (39)	2 (5)	16 (39)	<0.001 * 0.589 ^#^
Positive Euroimmun EUROLINE–WB	IgG	27 (68)	3 (8)	34 (83)	<0.001 * 0.176 ^#^
	IgM	18 (45)	2 (5)	15 (37)	<0.001 * 0.326 ^#^

Data are presented as numbers and (percentages). Statistical analysis used was Pearson’s Chi square test; * = *p*-value difference between PSL and HC, # = *p*-value difference between PSL and GP. PSL—patient group with persistent symptoms attributed to Lyme disease; HC—healthy controls; GP—patients in primary care.

## Data Availability

The datasets used can be accessed by sending a request to foekje.stelma@radboudumc.nl.

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
