# Peer review of "Classical Borrelia Serology Does Not Aid in the Diagnosis of Persistent Symptoms Attributed to Lyme Borreliosis: A Retrospective Cohort Study"

_life, 2023, doi:10.3390/life13051134_

Round 1
Reviewer 1 Report
Paper received by me for review by Foekje F. Stelma at al. presents a comparison of the results of serological tests performed in 4 groups of patients (including 1 group taken from the references and one control group) with various symptoms of Lyme borreliosis. As the authors themselves note, serological tests from different manufacturers confirm high variability of results, which is also confirmed by other researchers in their publications and the data in this paper do not provide enough evidence to draw conclusions about the value of the tests in clinical practice. Proper estimation of the sensitivity and specificity of tests used in practice requires well-designed cross-sectional studies conducted in appropriate clinical patient populations including patients from different geographical locations.
I found many flaws that should be revised.
L2 – I would suggest changing to Lyme borreliosis (or Lyme disease) instead of Lyme in the title L13 – I recommend changing the “Borrelia bacterium” to Borrelia spirochetes L19 – please rewrite that sentence so You don't use the word "persisting" twice L21- it looks like two spaces, please check the whole ms for similar flaws e.g. 24, 113, 203 L22 – please change "This study" to "Our study" or rewrite this sentence so that it does not repeat the previous sentence (L20) L30 – I recommend clarifying abbreviations, even common ones like ELISA, on first use and checking the entire manuscript for similar doubts, as some abbreviations and symbols (e.g. PTLDS, BmpA, Flagellar, VlsE, and many more) are used throughout the text, e.g. in the introduction, without explaining the meaning or just the description that it is a Flagellar antigen and not just a Flagellar (L109), and only at the end of the text in the discussion are some full names given L41- this sentence is not clear, please rewrite it a bit L53 – the instead of the L54-55 – firstly you mention Borrelia burgdorferii sensu stricto for the first time, it should be given with the full "sensu stricto", secondly B. burgdorfeii s.s. is not part of the B. burgdorfeii sensu lato complex, correct this sentence L67 - like I wrote in L30 L76 – Borrelia spp. instead of Borrelia spp., the same L231 L77- you can just write LB or Lyme so you don't repeat Lyme boreliosis all the time L80 - please correct the name of Borrelia in the entire manuscript, because once it is correctly written in italics and capital letter "Borrelia", another capital letter but not italic "Borrelia", and another lowercase letter without italics "borrelia", reading the text, I have the impression that each part was written by a different person in a little bit different, characteristic style and using a slightly different nomenclature, and there was no common unification of the text L98 – “studies can’t speak”, I know what do you mean but please rewrite this sentence L103 – should be so called “window period” L108 – full stop in a wrong place L109 – as suggested in L30 L110 - 22% of what? Clarify this sentence please L125 – If you cited a reference using the author's name, insert the reference number immediately after the name, e.g. Branda et al.[30]suggested…. L140 – where do you mention about third stage of borreliosis??? in the previous section you write about stage one and two only L163 – this sentence is not clear – rewrite it L170 – this sentence is not grammatically correct, rewrite it L180 – I supposed it should be n=40 not n=4- L199 – equivocal or borderline, equivocal and borderline instead of equivocal/borderline L216 – coefficient instead of coëfficient L256 – space between with and 37%, remove ). L264 – remove is Table 4 – I strongly recommend changing the colors in this table as it looks like a black and white copy, the same Figure 1, 2 and 3 L287 – what is 1,035? L311-313 – this sentence is not grammatically correct, rewrite it L375 – space after OspA L402-404 – this is your second hypothesis and it is placed in the discussion section, remove it from here and decide which hypothesis you are giving and if it corresponds to your results L430 – but one – which one? rewrite this section L460 - was the approval of the ethics committee valid? It was released in 2010 Please correct all results and numbers (especially in tables) given with commas (,) only some were given in the correct format (with dot .)!!! I strongly recommend reviewing the entire manuscript for many editorial flaws as well as the English language.
Author Response
Dear Reviewer,
Thank you for extensively reviewing our paper. We very much appreciate this and have applied most of your suggestions which improves the paper. Also, we will submit the paper for an official language check.
We would like to start by answering you first comment above, concerning the aim of this paper. You correctly state that this investigation does not provide enough evidence to draw conclusions about the value of the tests in clinical practice. However, the aim of this investigation was not to validate tests from various manufacturers. This has indeed been done by many others before us. The aim of this paper was to specifically compare the reactivity of serological tests between patient groups with well-defined Lyme borreliosis (our GP and EM groups) and patients with chronic and less defined symptoms that often are attributed to Lyme borreliosis (our PSL group). This is an interesting observation which has not had much attention earlier. Serology is often used to add to the proof that the symptoms are caused by Lyme borreliosis. Our main conclusion is that serology does not add to the diagnosis in Lyme borreliosis in the patient group with chronic symptoms. Further, our introduction, which was extended on request of the editor, gives a mini-review of various aspect of serology in Lyme borreliosis.
In the following you will find that we have addressed your comments point by point.
Yours sincerely,
Foekje F. Stelma
Reviewer 1
Paper received by me for review by Foekje F. Stelma at al. presents a comparison of the results of serological tests performed in 4 groups of patients (including 1 group taken from the references and one control group) with various symptoms of Lyme borreliosis. As the authors themselves note, serological tests from different manufacturers confirm high variability of results, which is also confirmed by other researchers in their publications and the data in this paper do not provide enough evidence to draw conclusions about the value of the tests in clinical practice. Proper estimation of the sensitivity and specificity of tests used in practice requires well-designed cross-sectional studies conducted in appropriate clinical patient populations including patients from different geographical locations.
L2 – I would suggest changing to Lyme borreliosis (or Lyme disease) instead of Lyme in the title
We have changed Lyme to Lyme borreliosis as suggested.
L13 – I recommend changing the “Borrelia bacterium” to Borrelia spirochetes
This part is the simple summary that should be easy to understand for layman. We agree that spirochetes are the correct biomedical terminology, however, as this has been written for laymen, we are in favor of using bacterium.
L19 – please rewrite that sentence so You don't use the word "persisting" twice
This writing is the result of the definition of our main research group (patients with persisting symptoms attributed to Lyme borreliosis (PSL)). We have now rewritten it to:
Blood tests are not validated in patients with persisting symptoms and doctors experience difficulties when testing for the cause of these continuing symptoms. This study compared blood tests in different groups of borrelia patients with patients having persisting symptoms after antibiotic treatment.
L21- it looks like two spaces, please check the whole ms for similar flaws e.g. 24, 113, 203
These spaces have been removed.
L22 – please change "This study" to "Our study" or rewrite this sentence so that it does not repeat the previous sentence (L20)
We changed it into:
This study compared blood tests in different groups of borreliosis patients with patients experiencing persisting symptoms after antibiotic treatment. Further we aimed to determine whether different patterns of test reactivity could be observed between groups.
L30 – I recommend clarifying abbreviations, even common ones like ELISA, on first use and checking the entire manuscript for similar doubts, as some abbreviations and symbols (e.g. PTLDS, BmpA, Flagellar, VlsE, and many more) are used throughout the text, e.g. in the introduction, without explaining the meaning or just the description that it is a Flagellar antigen and not just a Flagellar (L109), and only at the end of the text in the discussion are some full names given L41- this sentence is not clear, please rewrite it a bit
This is an unusual request as many of these abbreviations probably are known in the field, but we have decided to follow-up on this request, but only in the main text, as we are limited by words in the abstract. The abbreviations clarified are:
post treatment Lyme disease syndrome (PTLDS) L72
chemokine ligand 13 (CXCL13) L78
enzyme-linked immunosorbent assay (ELISA) L85
outer surface protein C (OspC) L114
basic membrane protein A (BmpA) L115
Flagellin protein L115
surface lipoprotein E (VlsE) L117
decorin binding protein A (p18 or DbpA) L270
p83/p100 protein L271
p39 protein p58 protein L272
(also called outer surface protein A (OspA)) L307
p41 flagellin antigen L378
We have rewritten L396-399:
….protein p31 (OspA) antigen, especially in the case of IgG testing, does not differentiate between patients with symptoms attributed to Borrelia infection (PSL) and healthy controls (HC; figure 1) . OspA also known as OspA…….
the Persistent Lyme Empiric Antibiotic Study Europe study (PLEASE-study) L184-185
Serion IgG and IgM ELISA (Virion/Serion GmbH, Würzburg, Germany) L195-196
Euroimmun EUROLINE-WB (Euroimmun Medizinische Labordiagnostika AG, Lübeck, Germany) L196
ELFA (Vidas, BioMerieux, Boston, MA, US) L205
ELISA (Euroimmun Medizinische Labordiagnostika AG, Lübeck, Germany) L206
Euroimmun EUROLINE-RN-AT (Euroimmun Medizinische Labordiagnostika AG, Lübeck, Germany). L209
The short form health survey (SF-36) L219
*Mikrogen RecomLine and Recombead (Mikrogen GmbH, Neuried, Germany) $ Serion Multianalyte (Virion/Serion GmbH, Würzburg, Germany). L239
L53 – the instead of the L54-55 – firstly you mention Borrelia burgdorferii sensu stricto for the first time, it should be given with the full "sensu stricto", secondly B. burgdorferi s.s. is not part of the B. burgdorferi sensu lato complex, correct this sentence
We have added sensu stricto where appropriate. However, also after consulting the literature, we still have the impression that Borrelia burgdorferi ss is part of the Borrelia burgdorferi sl complex, as are Borrelia garinii and Borrelia afzelii. (see also Mongodin et al. BMC Genomics 2013, 14:693). So we have not changed the text further.
L67 - like I wrote in L30 L76 – Borrelia spp. instead of Borrelia spp., the same L231 L77- you can just write LB or Lyme so you don't repeat Lyme borreliosis all the time
I have added the abbreviation LB in L53 and also changed Lyme borreliosis to LB in the rest of the, except for in titles, tables and figures.
L80 - please correct the name of Borrelia in the entire manuscript, because once it is correctly written in italics and capital letter "Borrelia", another capital letter but not italic "Borrelia", and another lowercase letter without italics "borrelia", reading the text, I have the impression that each part was written by a different person in a little bit different, characteristic style and using a slightly different nomenclature, and there was no common unification of the text
All has been improved to Borrelia.
L98 – “studies can’t speak”, I know what do you mean but please rewrite this sentence
We have rewritten this sentence to… some scientists define symptoms that develop within 10 weeks after exposure also to be early borreliosis.
L103 – should be so called “window period”
We have added so called…
L108 – full stop in a wrong place
We do not exactly see what you mean, but have tried to improve the sentence to be clearer: In EM the two tier serological approach was shown to have a limited sensitivity of 31-50%. [3, 29, 30] The early response is predominantly an IgM reaction with reactivity against….
L109 – as suggested in L30
Please see above.
L110 - 22% of what? Clarify this sentence please
We have rewritten the sentence to: Only 22% of serum samples in early localized disease is shown to be reactive in the combination of IgG ELISA….
L125 – If you cited a reference using the author's name, insert the reference number immediately after the name, e.g. Branda et al.[30]suggested….
We have corrected this as suggested.
L140 – where do you mention about third stage of borreliosis??? in the previous section you write about stage one and two only
We have added stage 3 (L147) to make it more clear: Late disseminated borreliosis (stage 3), present by ACA, Lyme arthritis and/or continuing neurological disease activity lasting more than 6 months[12, 30].
L163 – this sentence is not clear – rewrite it
We have rewritten it to: Considering the long-term Borrelia serologic memory after infection, it is not likely that the sero-reactive pattern of either IgG and IgM immune responses, will change over time. Therefore we conclude that serology does not contribute to the diagnostic work up in PSL.
Following a request of reviewer 2, proposing a more profound discussion on the value of serological diagnostics in PSL, we moved this whole paragraph (L146-165) to the end of the discussion.
L170 – this sentence is not grammatically correct, rewrite it
We have rewritten this sentence to: Finally, we investigated if pre-treatment serology could be used to predict which PSL patient was going to benefit from antibiotic treatment or not.
L180 – I supposed it should be n=40 not n=4-
This has been corrected
L199 – equivocal or borderline, equivocal and borderline instead of equivocal/borderline
This has been corrected
L216 – coefficient instead of coëfficient
This has been corrected
L256 – space between with and 37%, remove ).
This has been corrected
L264 – remove is Table 4 – I strongly recommend changing the colors in this table as it looks like a black and white copy, the same Figure 1, 2 and 3
I will discuss with the Journal what is possible.
L287 – what is 1,035?
This is the Chi square. I have reduced the number of decimals to 1, Chi square=1,0. However, as stated just before, this was not statistically significant.
L311-313 – this sentence is not grammatically correct, rewrite it
Rewritten to: . Pre-treatment Borrelia serology in patients with persistent symptoms attributed to Lyme borreliosis (PSL); comparison between patients comparison between patients with and without response to antibiotic treatment.
L375 – space after OspA
This has been corrected
L402-404 – this is your second hypothesis and it is placed in the discussion section, remove it from here and decide which hypothesis you are giving and if it corresponds to your results
Rewritten to: We observed a decreased frequency of VlsE-reactivity in our PSL population which may reflect IgG waning immune responses over time[39].
L430 – but one – which one? rewrite this section
We have deleted the sentence. This sentence is speculative.
L460 - was the approval of the ethics committee valid? It was released in 2010
Yes, we went to great lengths to get this approval, and the official papers are present at the Radboudumc Valorization office.
Please correct all results and numbers (especially in tables) given with commas (,) only some were given in the correct format (with dot .)!!! I strongly recommend reviewing the entire manuscript for many editorial flaws as well as the English language.
All the commas have been replaced by dots.
Reviewer 2 Report
In this study the authors compared blood tests in different groups of borrelia patients with patients persisting symptoms after antibiotic treatment, and aimed to determine whether different patterns of test reactivity between groups. They found that patterns of test positivity and reactivity did not differ between groups. However, various commercial test did show changing test results in the same person. Therefore, although patients present with persisting symptoms after treatment for Lyme borreliosis, testing is not useful, as treatments do not change the positive test result.
The study raises an important diagnostic result, however the authors should discuss and further stress the novel findings of their study compared with previous literuatre data.
Author Response
Dear reviewer,
Thank you for reviewing our paper and for recognizing that this paper raises a question with an important result. You ask us to further discuss the novelty of the finding in comparison to previous literature. To our knowledge there is hardly any literature on this exact topic and no studies have been performed to establish the value of serological diagnostics in patients with persisting symptoms after treatment for Lyme borreliosis. A paragraph reviewing the very little evidence on this subject was added in to the introduction (L146-165). We moved this paragraph to the discussion, as this highlights better the main message in this paper.
Yours sincerely,
Foekje F. Stelma
Reviewer 2
In this study the authors compared blood tests in different groups of borrelia patients with patients persisting symptoms after antibiotic treatment, and aimed to determine whether different patterns of test reactivity between groups. They found that patterns of test positivity and reactivity did not differ between groups. However, various commercial test did show changing test results in the same person. Therefore, although patients present with persisting symptoms after treatment for Lyme borreliosis, testing is not useful, as treatments do not change the positive test result.
The study raises an important diagnostic result, however the authors should discuss and further stress the novel findings of their study compared with previous literature data.
Reviewer 3 Report
This research provides a valuable comparison between serological tests for Lyme borreliosis from a variety of manufacturers and assesses whether serological testing has value for differentiating between early and late stages of the disease. The results indicate that serology is unable to accurately distinguish early and persistent forms, and is not very useful for diagnosis of, or predicting treatment success for, patients with PSL. The research also highlights variability between different manufacturers' tests and in the usefulness of the different antigens used in these assays. Overall, this paper is valuable in studying the diagnosis of Borreliosis and highlights the ongoing difficulties in Lyme diagnosis, particularly in differentiating the post-treatment forms of the disease.
The paper is mostly well written and well explained, although there are a few areas where the language needs to be tidied up to improve clarity. It is notable that a comparison of serology in treated/untreated PSL patients is not included, as this would be interesting. Minor edits are suggested below.
throughout - Borrelia should be written with a capital 'B'
line 14: "...by a bite of an infected tick"
line 21: "...tests in different groups of borreliosis patients with patients experiencing persisting symptoms..."
line 67: define PTLDS
line 109: correct to "flagellin" not flagellar
line 110: only 22% of what? patients?
line 132: correct to "Late disseminated borreliosis presents as ACA, Lyme arthritis..."
line 171 & line 438: change "on forehand" to "beforehand"
line 180: typo here, should be n=40
Table 1: in country row, spelling Romania
Table 3: it would be beneficial for interpreting the table to highlight the significant p values in the table with bold text or *, for example.
line 269: check the grammar of this sentence.
Table 5: In the table footnote it says # indicates significant difference between PSL and GP, but none of the p-values are significant.
line 285-286: In the graph, Figure 1A, the values of IgG-p31 are not the same, but in the text it says that they are both 18.
line 296: this heading should have a number 3.4
line 311: this heading's number should be changed to 3.5. The second part of the heading should be corrected to "... comparison of patients with or without response to antibiotic treatment"
What about post-treatment? were there any serological differences in treated vs untreated groups? This would be interesting and important to know.
Discussion
line 343: change "a-specific" to "non-specific"
lines 386-387: these should be "genotype" rather than "type"
line 430: different directions.., such as? Authors could give examples of these possible different directions.
Author Response
Dear reviewer,
We would like to thank you for your extensive reading of our paper. We have addressed most of the edits and responded to some of your questions. Also we will submit the manuscript for English language editing.
In the following you will find that we have addressed your comments point by point.
Yours sincerely,
Foekje F. Stelma
Reviewer 3
This research provides a valuable comparison between serological tests for Lyme borreliosis from a variety of manufacturers and assesses whether serological testing has value for differentiating between early and late stages of the disease. The results indicate that serology is unable to accurately distinguish early and persistent forms, and is not very useful for diagnosis of, or predicting treatment success for, patients with PSL. The research also highlights variability between different manufacturers' tests and in the usefulness of the different antigens used in these assays. Overall, this paper is valuable in studying the diagnosis of Borreliosis and highlights the ongoing difficulties in Lyme diagnosis, particularly in differentiating the post-treatment forms of the disease.
The paper is mostly well written and well explained, although there are a few areas where the language needs to be tidied up to improve clarity. It is notable that a comparison of serology in treated/untreated PSL patients is not included, as this would be interesting. Minor edits are suggested below.
throughout - Borrelia should be written with a capital 'B'
Borrelia has been rewritten to capital B and italic, as also requested by reviewer 1.
line 14: "...by a bite of an infected tick"
This has been rewritten accordingly.
Line 21: “…tests in different groups of borreliosis patients with patients experiencing persisting symptoms”.."
This has been rewritten to: This study compared blood tests in different groups of borreliosis patients with patients experiencing persisting symptoms after antibiotic treatment.
line 67: define PTLDS
This has been rewritten to: Considering Borrelia diagnostics, post treatment Lyme disease syndrome (PTLDS)…..
line 109: correct “o "flagel”in" not flagellar
This has been corrected as requested.
line 110: only 22% of what? patients?
We have clarified this sentence as follows: Only 22% of serum samples in early localized disease is shown to be reactive…….
line 132: correct to "Late disseminated borreliosis presents as ACA, Lyme arthritis..."
This has been corrected as requested: Late disseminated borreliosis (stage 3), present as ACA, Lyme arthritis and/or continuing neurological disease
line 171 & line 438: change "on forehand" to "beforehand"
The first sentence was rewritten to: Finally, we investigated if pre-treatment serology could be used to predict which PSL patient was going to benefit from antibiotic treatment or not.
The second sentence was rewritten to: Further it is not possible to distinguish beforehand between patients who will or will not respond to antibiotic treatment by serological testing
line 180: typo here, should be n=40
This has been corrected
Table 1: in country row, spelling Romania
This has been corrected as requested.
Table 3: it would be beneficial for interpreting the table to highlight the significant p values in the table with bold text or *, for example.
Table 3 shows a column with p-values for each comparison. The lower the p-value, the more convincing the difference between the groups is. We feel that it is the pattern of differences between the two groups and the magnitude of the p-value which provides an impression of what to expect when performing serology in those two groups. We are not convinced that adding highlights or * will make the data clearer.
line 269: check the grammar of this sentence.
This sentence was rewritten to: Samples from PSL, HC and GP were tested using similar manufacturers (table 1) and therefore comparable.
Table 5: In the table footnote it says # indicates significant difference between PSL and GP, but none of the p-values are significant.
The * and # indicates which groups have been compared. This sentence has been rewritten to: * = p-value difference between PSL and HC, # = p-value difference between PSL and GP.
line 285-286: In the graph, Figure 1A, the values of IgG-p31 are not the same, but in the text it says that they are both 18.
We have checked the data in SPSS. There has been an omission in the text. The correct figures are: ….outer surface protein A (OspA) was seen more frequently in HC 7/40 (18%) and GP 8/40 (20%) when compared to PSL 4/40 (10%; Chi square=1.0; p=0.309)…..
line 296: this heading should have a number 3.4
This has been corrected.
line 311: this heading's number should be changed to 3.5. The second part of the heading should be corrected to "... comparison of patients with or without response to antibiotic treatment"
This all has been corrected to; 3.5. Pre-treatment Borrelia serology in patients with persistent symptoms attributed to Lyme borreliosis (PSL); comparison between patients with or without response to antibiotic treatment.
What about post-treatment? were there any serological differences in treated vs untreated groups? This would be interesting and important to know.
Unfortunately sampling of serum after treatment was not included in the protocol. Therefore we are not able to study the serology after treatment and perform a groups comparison at this time point.
Discussion
line 343: change "a-specific" to "non-specific"
We have corrected this as requested.
lines 386-387: these should be "genotype" rather than "type"
We have corrected this as requested.
line 430: different directions.., such as? Authors could give examples of these possible different directions.
We have deleted this sentence as this sentence is speculative.
Round 2
Reviewer 1 Report
I accept the corrections made in the manuscript, explanations of minor uncertainties are given below:
“ L108 – full stop in a wrong place”, now L113 as shown below
“We do not exactly see what you mean…” – (was has been shown to have a limited sensitivity of 31-50%. [3, 29, 30] This The 113)
I meant full stop after 31-50% - it should be after brackets.
“All the commas have been replaced by dots.”
Please replace the commas with dots in the supplementary file as well.
“We have added sensu stricto where appropriate. However, also after consulting the literature, we still have the impression that Borrelia burgdorferi ss is part of the Borrelia burgdorferi sl complex, as are Borrelia garinii and Borrelia afzelii. (see also Mongodin et al. BMC Genomics 2013, 14:693). So we have not changed the text further.”
You're absolutely right, sorry for that comment. Please correct only sensu stricto (ss or s.s.) it should be without italics.
Author Response
Dear Reviewer 1,
Thank you for being so thorough. We have complied to correction 1 and 3.We do not quite understand the comment about the supplementary file, as there is none. But I saw that I had not updated the original tables files. This has been done now. See for the details hereunder.
Yours sincerely,
Foekje Stelma
--------
I accept the corrections made in the manuscript, explanations of minor uncertainties are given below:
“ L108 – full stop in a wrong place”, now L113 as shown below
“We do not exactly see what you mean…” – (was has been shown to have a limited sensitivity of 31-50%. [3, 29, 30] This The 113)
I meant full stop after 31-50% - it should be after brackets.
- The full stop has been moved to after the brackets.
“All the commas have been replaced by dots.”
Please replace the commas with dots in the supplementary file as well.
- We do not have supplementary files. I updated the original tables files.
“We have added sensu stricto where appropriate. However, also after consulting the literature, we still have the impression that Borrelia burgdorferi ss is part of the Borrelia burgdorferi sl complex, as are Borrelia garinii and Borrelia afzelii. (see also Mongodin et al. BMC Genomics 2013, 14:693). So we have not changed the text further.”
You're absolutely right, sorry for that comment. Please correct only sensu stricto (ss or s.s.) it should be without italics.
- The italics in sensu stricto, sensu latu and ss has been removed.
Reviewer 2 Report
The authors addressed the raised points.
Author Response
There are no points to be adressed. Thank you.